# Diagnostic performance and usability of the VISITECT CD4 semi-quantitative test for advanced HIV disease screening

Zibusiso Ndlovu[1]*, Lamin Massaquoi[2], Ndim Eugene Bangwen[2], John N. Batumba[2], Rachelle U. Bora[2], Joelle Mbuaya[2], Roger Nzadi[2], Nadine Ntabugi[2], Patrick Kisaka[2], Gisele Manciya[2], Ramzia Moudashirou[3], Harry Pangani[4], Patrick Mangochi[4], Roberta Makoko[4], David Van Laeken[4], Claude Kwitonda[4], Yuster Ronoh[5], Kuziwa Kuwenyi[5], Reinaldo Ortuno[5], Douglas Mangwanya[6], Edmore Zvidzai[6], Tapiwa Mupepe[6], Sekesai Zinyowera[7], Emmanuel Fajardo[8], Tom Ellman[1]

1 Southern Africa Medical Unit, Médecins Sans Frontiéres, Cape Town, South Africa, 2 Médecins Sans Frontiéres, Kinshasa, Democratic Republic of Congo, 3 Médecins Sans Frontières, Conakry, Guneia, 4 Médecins Sans Frontières, Nsanje, Malawi, 5 Médecins Sans Frontières, Gutu, Zimbabwe, 6 Ministry of Health and Child Care, Harare, Zimbabwe, 7 National Microbiology Reference Laboratory, Ministry of Health and Child Care, Harare, Zimbabwe, 8 Médecins Sans Frontières, Barcelona, Spain

* ndlovinizee@gmail.com

**Data Availability Statement:** Data are available via Dryad (doi: 10.5061/dryad.fbg79cnrg).

## Abstract

### Background

In sub-Saharan Africa, a third of people starting antiretroviral therapy and majority of patients returning to HIV-care after disengagement, present with advanced HIV disease (ADH), and are at high risk of mortality. Simplified and more affordable point-of-care (POC) diagnostics are required to increase access to prompt CD4 cell count screening for ambulatory and asymptomatic patients. The Visitect CD4 Lateral Flow Assay (LFA) is a disposable POC test, providing a visually interpreted result of above or below 200 CD4cells/mm³. This study evaluated the diagnostic performance of this index test.

### Methods

Consenting patients above 18years of age and eligible for CD4 testing were enrolled in Nsanje district hospital (Malawi), Gutu mission hospital (Zimbabwe) and Centre hopitalier de Kabinda (DRC). A total of 708 venous blood samples were tested in the index test and in the BD FACSCount assay (reference test method) in the laboratories (Phase 1) to determine diagnostic accuracy. A total of 433 finger-prick (FP) samples were tested on the index test at POC by clinicians (Phase 2) and a self-completed questionnaire was administered to all testers to explore usability of the index test.

### Results

Among 708 patients, 67.2% were female and median CD4 was 297cells/mm³. The sensitivity of the Visitect CD4 LFA using venous blood in the laboratory was 95.0% [95% CI: 91.3–97.5] and specificity was 81.9% [95% CI: 78.2–85.2%]. Using FP samples, the sensitivity of

**Funding:** The author(s) received no specific funding for this work.

**Competing interests:** The authors have declared that no competing interests exist.

the Visitect CD4 LFA was 98.3% [95% CI: 95.0–99.6] and specificity was 77.2% [95% CI: 71.6–82.2%]. Usability of the Visitect CD4 LFA was high across the study sites with 97% successfully completed tests. Due to the required specific multiple incubation and procedural steps during the Visitect CD4 LFA testing, few health workers (7/26) were not confident to manage testing whilst multi-tasking in their clinical work.

## Conclusions

Visitect CD4 LFA is a promising test for decentralized CD4 screening in resource-limited settings, without access to CD4 testing and and it can trigger prompt management of patients with AHD. Lay health cadres should be considered to conduct Visitect CD4 LFA testing in PHCs as well as coordinating all other POC quality assurance.

## Introduction

CD4 cell count test has been used to determine when HIV infected patients should start antiretroviral therapy (ART) and to monitor ART [1–7]. CD4 cell count test has been key in predicting disease progression and death among people living with HIV, and it has been crucial for assessing eligibility of prophylaxis for opportunistic infections (OIs) [3, 4].

However, in 2013, World Health Organization (WHO) recommended the use of HIV Viral load (VL) for monitoring ART over CD4 cell counts as VL accurately detects virological failure before immunological or clinical deterioration [8]. As such, nearly 95% of low and middle income countries (LMIC) have implemented policies for scale up of VL testing [9].

In-spite of continuous improvements in ART coverage and monitoring, the decline in HIV/AIDS-related mortality is stalling. In 2018 nearly 770, 000 people died from AIDS related-diseases, a mere 30,000 less than the previous year [10]. Many HIV programs in LMICs continue to report significant proportions of treatment-naive and treatment-experienced patients presenting to care with advanced HIV disease (AHD). AHD is defined as an adult, adolescent, or child greater than 5 years old with a CD4 cell count <200 cells/mm$^3$ or a WHO clinical stage 3 and 4 event as well as all children less than 5 years [2]. Patients presenting with AHD are at high risk of death, even after starting ART. CD4 cell count testing is important to aid in identifying ambulatory and asymptomatic patients eligible for further AHD screening and a result of CD4 cell count less than 200cells/mm$^3$ triggers screening for urinary mycobacterial tuberculosis lipoarabinomannan antigen (TB LAM) and cryptococcal antigen (CrAg) [2] using TB LAM and CrAg POC lateral flow assays (LFA). This package of care for AHD should be offered at hospitals, decentralized primary health care clinics including at peripheral sites and even through mobile outreach [2].

Although there are at least three commercially available point-of-care (POC) or near-POC CD4 assays in the market (Abbott Pima (Abbott, Chicago, IL, USA), BD FACSPresto (BD Biosciences, San Jose, CA, USA) and CyFlow miniPOC (Sysmex Partec, Goertlitz, Germany)), these rely on instrument analysers for CD4 measurement [11]. They require a considerable initial capital investment on instrumentation, usually more than USD $5,000, as well as a continuous service and maintenance for instrument repairs. It has also been previously reported that instrument-based POC CD4 technologies are prone to breakdowns and can generate a considerable amount of invalid test results [12] therefore adding to the test cost. Furthermore, with focus on VL scale-up to monitor ART, and stagnating donor health funding [13], sustainability of instrument-based CD4 technologies is uncertain. As such, there is a need for instrument-

free and affordable POC CD4 technologies amenable for decentralization to expedite implementation of the AHD package which has a mortality benefit.

Developed by the Burnet Institute, the Omega VISITECT CD4 Advanced Disease Lateral Flow Assay (Visitect CD4 LFA) (Omega Diagnostics, Scotland, UK), is a disposable POC test that offers an estimation of the CD4 protein on the surface of CD4+ T cells, providing a semi-quantitative results at a threshold of 200cells/mm$^3$ [14]. This study sought to evaluate the diagnostic accuracy and feasibility-of-use for this first-ever instrument free POC CD4 test, which provides a visually interpreted result, of above or below 200 CD4 cells/mm$^3$, after 40 minutes.

## Methods

### Study design and setting

This was a prospective two phased diagnostic accuracy and feasibility-of-use evaluation study, conducted in Médecins Sans Frontiéres (MSF) supported health facilities of; Nsanje district hospital (NDH) (Malawi), Gutu mission hospital (GMH) (Zimbabwe) and Centre hopitalier de Kabinda (CHK) Democratic Republic of Congo (DRC). CHK is a tertiary HIV referral health facility whilst NDH and GMH are rural district hospitals. Phase 1 of the study was a diagnostic accuracy evaluation of the Visitect CD4 LFA compared to the BD FACSCount assay (BD Biosciences, San Jose, CA, USA) within the study site laboratories while Phase 2 was carried out at the patient site, to assess the feasibility of use of the index test.

### Study population

All HIV positive adult (18 to 64years) patients eligible for a CD4 test for any reason and providing a written informed consent, were eligible.

### Sample size

Sample size estimation for Phase 1 was guided by an expected 25% prevalence of patients with CD4<200cells/mm$^3$ from the three study sites and expected point estimates of sensitivity and specificity of 80% and 75% respectively in the index test. A minimum of 660 patients were considered to achieve margins of sampling error of approximately 2% for point estimates with more than 90% power. For phase 2 study, sample size estimation was established on a convenient sampling of a minimum of 425 patients, based on pragmatic bi-monthly CD4 cell testing volumes in the study sites.

### Sample collection

In Phase 1, venous EDTA blood sample (3ml) was collected from each consenting patient for routine CD4 testing at the laboratory (on the BD FACScount platform) and the excess sample was also tested for CD4 cell count in the index Visitect CD4 LFA (Fig 1 flow chart). During Phase 2, all prospective eligible patients who provided written informed consent, had routine venous EDTA blood sample collected together with a finger-prick (FP) sample. The FP sample was for simultaneous POC testing, on the Visitect CD4 LFA and on the PIMA CD4 POC, whereas the EDTA sample was sent for testing at the laboratory.

### Testing procedures

BD FACSCount CD4 cell count: EDTA whole blood samples were transported at room temperature to the facility laboratory within 2 hours of collection. The samples were processed in the BD FACSCount CD4 machine (Becton Dickinson, San Jose, California, USA) according to the manufacturers' instructions, and laboratory standard operating procedures (SOPs) [15].

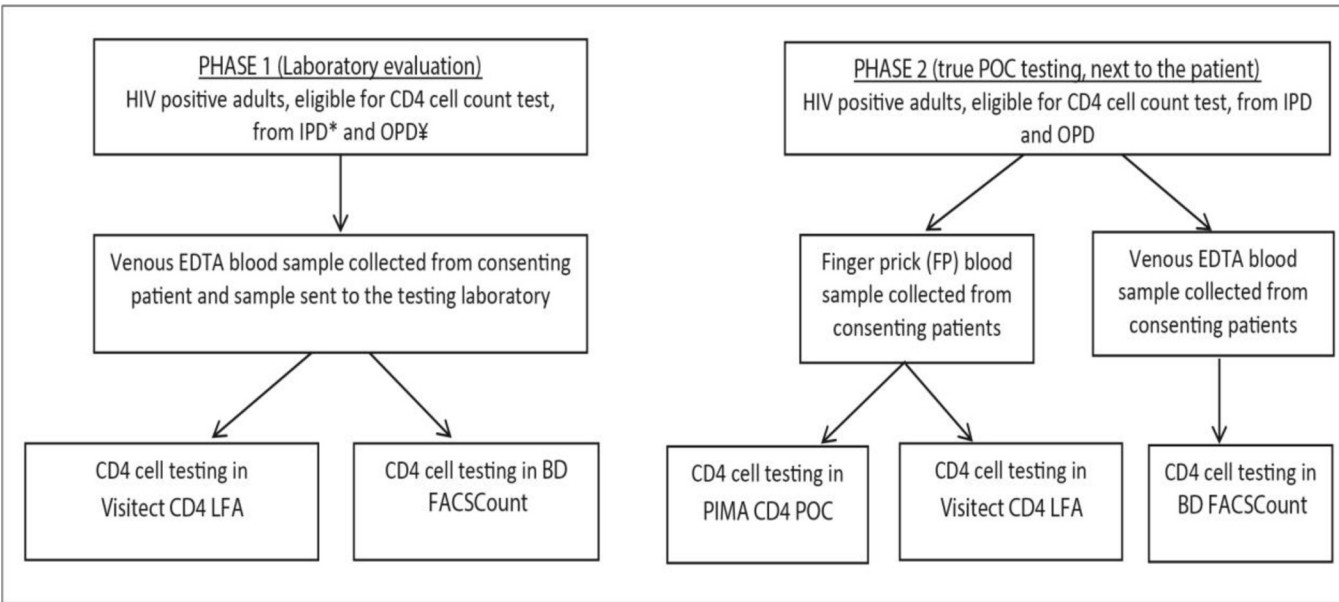

**Fig 1. Study sample collection and testing flow.** Key: *IPD-in-patient department, ¥OPD-out patient department.

For PIMA CD4 POC testing; the finger prick (FP) blood sample was added to the Pima CD4 reagent cartridge and the cartridge was inserted into the Abbott PIMA CD4 device (*formerly Alere*) and processed following manufacture instructions and laboratory SOPs [16]. Commercial controls and proficiency testing (PT) results were used to assure quality in CD4 testing services.

The Visitect CD4 LFA test kit contains a disposable test device, one buffer bottle, a 30μL micropipette, a safety lancet and an alcohol swab. User training was provided by the Omega team. The Visitect CD4 LFA testing was conducted following the manufacturer's instructions for use (IFU); briefly, FP blood was obtained using the safety lancet; and the disposable pipette was used to collect 30μL of the FP sample. The blood sample was added into the Well A and incubated for 3 minutes, during which red blood cells and monocytes are retained in the blood sample collection pad (S1 Appendix). Thereafter, one drop of buffer is added into Well A and incubated for 17 minutes, during which other white blood cells (including CD4+ T-cells) migrate to a reaction area where cell lysis occurs resulting in the release of the full length CD4 for capture in the test strip. After 17 minutes, 3 drops of the buffer are then added into Well B so as to release the colloidal gold labeled monoclonal antibody conjugate that forms the reference line and the control line. The test results can be read after 20 minutes. Test results are visually interpreted by comparing the color intensity of the test line (T) with that of the reference line (200); if the test line (T) is darker than reference line (200), the patient has CD4 cell count >200cells/mm$^3$ and if the T line is fainter or similar intensity to the 200 reference line, then the patient has CD4cell count <200cells/mm$^3$ (S2 Appendix). Venous EDTA blood can also be used with the test. The Visitect CD4 LFA test kit can be stored within temperature ranges of 2–30˚C.

In the study, Visitect CD4 LFA results were read by two independent operators who were blinded to each other's result and to the BD FACScount absolute CD4 result.

During Phase 1, the study also assessed the precision of Visitect CD4 LFA test (repeatability of CD4 measurements from the same sample in same environmental conditions) by the same individual. Visitect CD4 LFA test result reproducibility was assessed through varying the incubation times to beyond manufacture IFU by different laboratory technicians. The study also investigated Visitect CD4 LFA test result stability; where device testing results were first read

following the manufacturer IFU, and later re-read by two blinded and independent readers daily for one week, to assess stability of results.

A self-completed questionnaire to explore Visitect CD4 LFA usability was administered to all testers who had conducted a minimum of 10 tests.

## Data management and ethical approval

Data was entered into a password-protected Microsoft Access data base and only accessible to authorised staff involved in data management. Diagnostic performance was explored using sensitivity, specificity and precision. Data was analysed in Stata-14. Usability responses from the testers' questionnaires were scored and described as percentages. This study was approved by the University of Malawi-College of Medicine Research and Ethics Committee (1891), University of Kinshasa Ethics Review Board (078/2018), Medical Research Council of Zimbabwe (MRCZ/A/2430) and by the MSF Ethical Review Board (1747).

# Results

## Characteristics of study participants in Phase 1

A total of 708 patients were enrolled into the diagnostic accuracy study (Phase 1) and 63.6% were from DRC. Overall median age was 42 years [IQR: 34–50] and 67.2% of the participants were female. Median CD4 cell count was 297 cells/mm$^3$ [IQR: 170–499] and 31.2% had CD4<200cells/mm$^3$, Table 1.

**Table 1. Characteristics of study participants in Phase 1.**

| Variable | Proportion n (%) |
|---|---|
| **Total** | 708 |
| **Gender** | |
| Male | 227 (32.1) |
| Female | 476 (67.2) |
| Unknown | 5 (0.7) |
| **Median age (years) [IQR]** | 42 [34–50] |
| **Median CD4 (cells/mm$^3$)** | 297 [170–499] |
| **Reason for CD4 testing** | |
| Advanced HIV disease screening | 272 (38.4) |
| ART initiation | 267 (37.7) |
| ART monitoring | 98 (13.8) |
| Other | 71 (10.0) |
| **Participant morbid conditions§** | |
| TB | 178 (25.1) |
| Malaria | 26 (3.7) |
| Cryptococcal meningitis | 14 (2.0) |
| Toxoplama ghondii | 7 (1.0) |
| Severe bacterial sepsis | 11 (1.6) |
| Other | 152 (21.5) |
| Missing | 320 (45.1) |

Key:

§ Majority of patients were presenting with multiple morbidities. Also, as data was collected prospectively, a significant number of patients had 'unknown diagnosis' upon time of initial consultation (i.e time of study patient data collection).

## Diagnostic performance of the Visitect CD4 LFA test compared to the BD FACScount

Compared to BD FACScount assay, the sensitivity of the Visitect CD4 LFA (at the 200cells/mm$^3$ threshold) using venous blood samples in the laboratory was 95.0% [95% CI: 91.3–97.5] and specificity was 81.9% [95% CI: 78.2–85.2%].

The median CD4 cell count of the 88/708 (Table 2) misclassified by Visitect CD4 LFA as CD4<200cells/mm$^3$ was 252cells/mm$^3$ [IQR: 222–306], whereas for the 11/708 misclassified as CD4>200cells/mm$^3$, their median CD4 cell count was 178cells/mm$^3$ [IQR: 134–185].

## Diagnostic performance of Visitect CD4 LFA at different CD4 cut-offs of the reference test

In restricting the analysis to samples with CD4<100cells/mm$^3$ cut-off in the reference test, the sensitivity of the Visitect CD4 LFA (at its standard 200cells/mm$^3$ threshold) in detecting samples with CD4<100cells/mm$^3$ was 98.1% [IQR: 93.5–99.8] and specificity was 68.0% [IQR: 64.1–71.7]. Table 3 below, shows further diagnostic performance of the Visitect CD4 LFA to different CD4 cut-offs ranges of the reference test.

**Visitect CD4 LFA precision testing data (intra-assay).**   Of the 12 excess EDTA samples (6 with CD4<200 and 6 with CD4>200cells/mm$^3$ on the reference test), tested 3 times each, all gave results concordant with the reference test (100% precision).

**Visitect CD4 LFA reproducibility data from varying incubation times.**   Of 6 EDTA samples tested 5 times each (three with true CD4>200 and three with <200) while varying only the first incubation times (to 1min, 2 mins, 5 mins, 8 mins and 10 mins); there was incidence of result transition among those with initial CD4>200 (5/30; 16.7%) to a result of CD4<200, if the first incubation times were less-than 3mins. However, when the first incubation time was greater than 3mins, only one test transitioned to CD4>200 (1/30; 3.3%) from true CD4<200.

Of another 6 EDTA samples tested 5 times each while varying only the second incubation times (to 5 mins, 10 mins, 20 mins, 25 mins and 30 mins); there was no overall result transition observed as test results remained similar. Similarly, there was no overall result transition observed while varying only the third incubation times (10 mins, 15 mins, 25 mins, 30 mins and 35 mins), as test results remained similar.

**Visitect CD4 LFA result re-reading stability data.**   After one-week of ambient laboratory temperature (24–28°C) incubation of the Visitect CD4 LFA test result cassettes, with double blinded daily result re-reading, there was no observed incidence of Visitect CD4 LFA result transition between states. Of the total 18 results (12 true CD4 greater than 200 cells/mm$^3$ and 6 true CD4 less than 200 cells/mm$^3$); all maintained their initial testing results. The control and reference lines remained fairly stable throughout the re-reading period.

**Table 2.  Contingency table of Visitect CD4 LFA test results with BD FACScount.**

|  | Visitect CD4 LFA | | |
|---|---|---|---|
|  | CD4<200 | CD4>200 | Total |
| BD FACScount |  |  |  |
| CD4<200 | 210 | 11 | 221 |
| CD4>200 | 88 | 399 | 487 |
| Total | 298 | 410 | 708 |

**Table 3. Diagnostic performance of Visitect CD4 LFA at different CD4 cut-offs of the reference test.**

| CD4 range (n; %) | Sensitivity (%), 95% CI | Specificity (%), 95% CI |
|---|---|---|
| <100 (108; 15.3%) | 98.1 [93.5–99.8] | 68.0 [64.1–71.7] |
| 100–200 (113; 16.0%) | 92.0 [85.4–96.3] | 67.4 [63.5–71.2] |
| <200 (221; 31.2%) | 95.0 [91.3–97.5] | 81.9 [78.2–85.2] |
| <350 (417; 58.9%) | 68.8 [64.1–76.7] | 96.2 [93.3–98.1] |

## Visitect CD4 LFA usability at point-of-care (Phase 2)

A total of 433 patients were enrolled into the feasibility evaluation of the study (Phase 2) and 343 (79.2%) were from DRC. Overall median age was 41 years [IQR: 33–48], 67% were female and median CD4 cell count was 242 cells/mm$^3$ [IQR: 103–436]. Compared to BD FACScount assay, the sensitivity of the Visitect CD4 LFA (at the 200cells/ul threshold) using finger prick blood samples was 98.3% [95% CI: 95.0–99.6], specificity was 77.2% [95% CI: 71.6–82.2%]. The kappa statistical agreement between PIMA CD4 and Visitect CD4 LFA testing was 85.2%.

Out of 26 health care workers (HCW) who performed the Visitect CD4 LFA testing (8 nurses, 9 doctors, 9 lab techs), all had conducted more than 10 un-assisted tests and overall testing success test rate was 97.2% (1141/1173) with only 32 invalids observed due to operator failures (buffer added into incorrect Well, reading test result after 20mins and adding sample into incorrect Well). In the hands of the testers (both laboratory and non-laboratory trained testers), the median hands-on time for a single Visitect CD4 LFA test was 3mins [2-6mins] and the overall average testing time (from commencement of FP sample collection to result interpretation) was 45 minutes. Ninety percent reported that it was easy to comprehend the instructions after training, and after conducting a median of 4 tests [IQR: 1–6], testers felt confident in independently conducting the Visitect CD4 LFA.

Visitect CD4 LFA test result interpretability was one of the highest risk stage for user-errors; where 10 (38.5%) were not confident to interpret test results by comparing it to the reference threshold 200 line only, but they needed the pictorial IFU result picture to help correctly distinguish test lines, especially for test lines (T) very close in intensity to the reference line. Due to the precise incubation times and the multi procedural steps required for performing the test, a few health workers (27%) were not confident to manage the Visitect CD4 LFA testing whilst multi-tasking in their other work roles, however, this was less-felt so by the laboratory based staff. On overall perception, 70% participants would use the test kit again, and 85% recommended its use to primary and peripheral health care levels.

## Discussion

The Visitect CD4 LFA achieved a high level result agreement with reference testing instruments when used by laboratory trained HCW on venous blood (sensitivity of 95.0% and specificity of 81.9%) and by clinicians on FP samples at POC (sensitivity 98.3% and specificity of 77.2%). The slight reduction in specificity when used by clinicians is possibly due to the use of FP samples at POC, as opposed to venous blood samples and testers generally considered FP sample collection to be slightly challenging, Fig 2. Studies have also found that FP sampling requires adequate training so as to minimize excessive finger squeezing during sample collection, as this can lessen the possible dilution effect of tissue fluid [17–19]. Nonetheless, with adequate training and regular FP sample collection, HCWs could achieve similar results with venous blood sample testing.

Data from this study showed that Visitect CD4 LFA has a trend of better sensitivity at very low (<100cells/mm$^3$) and better specificity at very high CD4 cell counts (>350cells/mm$^3$),

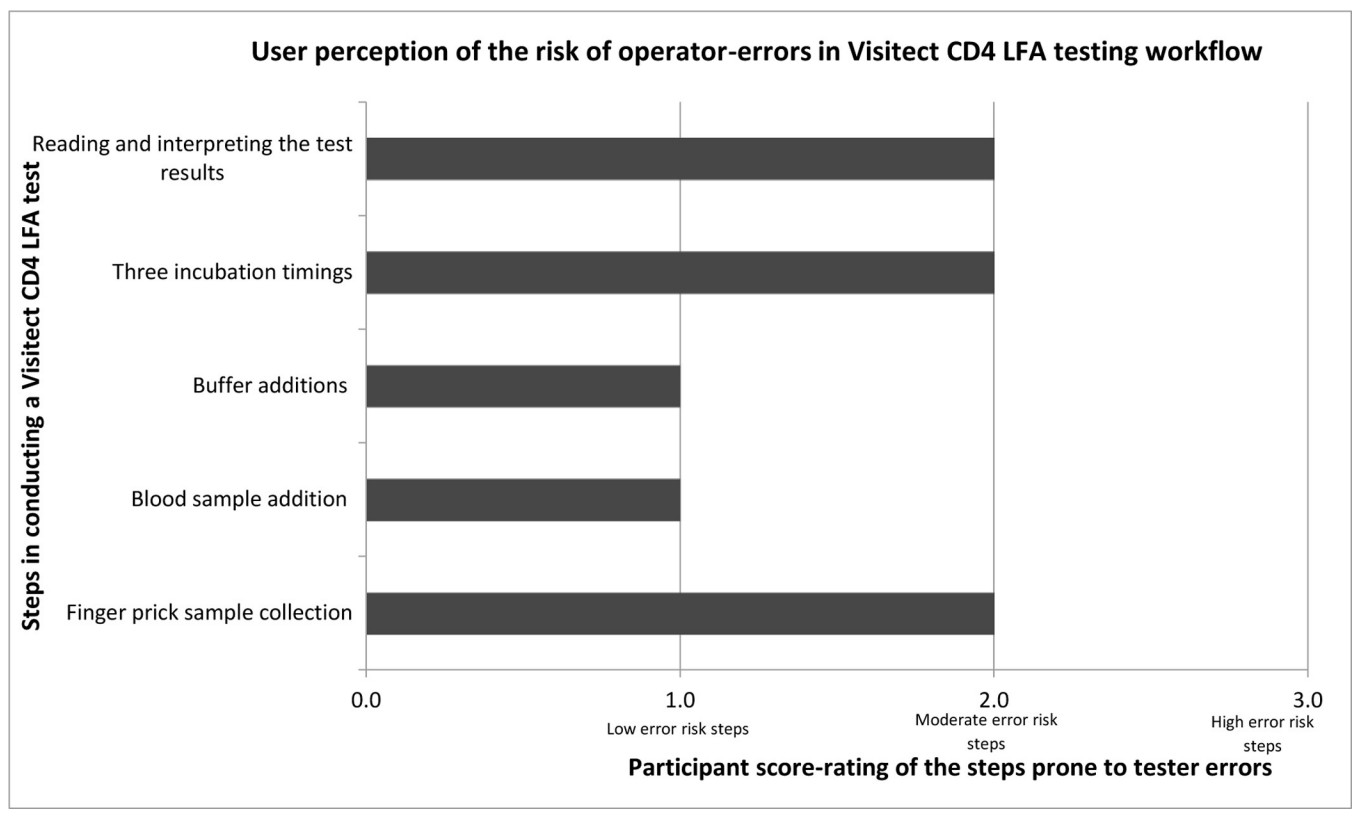

**Fig 2. Perception of the risk of operator-errors in the Visitect CD4 LFA testing workflow among testers.**

Table 3. This indicates the index test's reliability in detecting those at greatest risk of AHD (in need for CrAg and urine TB LAM LFA tests) and reliability in excluding those with higher CD4 categories (avoidance of unnecessary further tests for those at least risk). Studies have shown that majority of CM manifests at very low CD4 cell counts 100–150 cells/mm³ [20] and utility of urine TB LAM is best at CD4<200 [2], as such, Visitect CD4 LFA could expedite utilization of these minimal AHD diagnostic tools.

There was a significant proportion of patients who were misclassified by Visitect CD4 LFA (in Phase 1 and 2 of the study) as having CD4 less than 200 cells/mm³ (147/1141; 12.9%) and this could trigger potential 'over-use' of CrAg and urine TB LAM tests. These misclassified patients still had a relatively low median CD4 cell count (259 cells/mm³ [IQR: 224–315]). In ambulatory settings where median CD4 counts may be high, this lack of specificity will result in a relatively low positive predictive value. Nonetheless, patients with AHD have a very high mortality risk [20] and use of a simple and affordable screening test is justifiable even if it can result in few wasted investigations. On the other hand, only 14/1141 patients with CD4 cell counts less than 200 cells/mm³ were misclassified as not having AHD and they could have missed further management.

Varying the test incubation times produced a different mix of results and testers must adhere to all the three IFU stipulated precise sequential incubation times of 3 then 17 and 20 minutes. If the first incubation time is less than the manufacture stipulated 3 minutes, the test line (T) is likely to be faint (potentially over-diagnosing AHD), however, if the first incubation is beyond 3 minutes, possibly strong test lines (T) can be formed (potentially under-diagnosing AHD).

Even though delayed Visitect CD4 LFA result re-reading did not produce any result transitions, we do not advise such, either for monitoring or validating another testers' result interpretation.

Implementation of the Visitect CD4 LFA must be accompanied by clear examples of possible test result interpretations in the IFU as testers will struggle to interpret test (T) lines, especially test results line with intensity closer to the reference threshold. Additional support, such as provision of instructional video, is recommended. With adequate training of HCW and with many testing opportunities or requests, the test could be easy to implement in practice.

Visitect CD4 LFA should be used as a rule-out test and it can be used together with WHO symptom screen especially as it can identify asymptomatic AHD cases. Studies have shown that clinical screening alone could miss up-to 50% of AHD cases in ambulatory settings [20].

Where access to CD4 testing instruments is lacking, particularly at primary care, use of Visitect CD4 LFA may-be valuable to ensure early identification of those at risk of AHD and mortality. Such a test at peripheral clinics, could minimize referral of CD4 tests and referral of uncomplicated patient cases to higher-level clinical sites. Furthermore, mobile clinics, key populations (KP) and even community ART clubs may also benefit from use of Visitect CD4 LFA.

The burden of carrying out this and other essential simple tests [21, 22] may require investment in a dedicated lay cadre to ensure that the precise multi-stage manipulation of this test is accurately adhered to, plus reflex testing for samples with CD4 cell count less than 200cells/mm$^3$. This cadre could also coordinate all other POC quality assurance (QA) in the PHC facility. The much anticipated FujiFilm TB urine LAM POC test (FujiFilm, Tokyo, Japan) also has a multi-step testing procedure which requires precision and concentration during the incubation times [23]; as such, a specific lay cadre could streamline these and other tests for reliable and prompt accurate results. National programs already have different lay cadres who could be responsible and accountable for AHD at PHCs. For example, in Malawi, Health Diagnostic Assistants (HDAs), have solely been utilized by partners (MSF, Partners-In-Hope among others) for other activities including AHD POC testing at PHCs without laboratories. Other countries have lay counselors, nurse assistants, health facility navigators, community health workers, phlebotomists, all of whom can potentially take up this role.

It would be valuable if the Visitect CD4 LFA manufacturer considers revising the current test cassette and make an inscription of the blood and of a buffer drop next to Well A, whereas in Well B three buffer drops must be inscribed. This could minimize mistakes of testers adding blood samples or buffer drops to incorrect wells as it was noticed in the present study. We recommend pilot implementation of AHD package of Visitect CD4 LFA, together with CrAg and urine TB LAM, to explore their feasibility at peripheral settings including assessing patient's acceptability of such POC tests.

Strengths of this study include multi-country implementation and large sample sizes especially within cohort of patients with high immunosuppression (intended beneficiaries of such a test). The main limitation of this study is that it was only conducted at a tertiary and secondary health facility and not in peripheral PHCs.

## Conclusion and recommendation

Visitect CD4 LFA is an assuring test for decentralized CD4 testing in resource-limited settings, especially within PHCs with no access to CD4 testing instruments and it can trigger prompt management of patients with AHD. Lay health cadre roles should be reviewed with a consideration to assign Visitect CD4 LFA POC testing, including responsibility for QA of other POC tests in PHCs, to this cadre.

## Supporting information

**S1 Appendix. Visitect CD4 LFA testing procedure using finger prick blood sample.**
(TIF)

**S2 Appendix. Visitect CD4 LFA result interpretation.**
(TIF)

## Acknowledgments

We express our gratitude to health care workers in DRC, Zimbabwe and Malawi who conducted all the blood testing, and to patients who consented to participate in the study.

## Author Contributions

**Conceptualization:** Zibusiso Ndlovu, Emmanuel Fajardo, Tom Ellman.

**Data curation:** Zibusiso Ndlovu, Ramzia Moudashirou, Roberta Makoko, Claude Kwitonda.

**Formal analysis:** Zibusiso Ndlovu.

**Investigation:** Zibusiso Ndlovu, Roger Nzadi, Nadine Ntabugi, Patrick Kisaka, Gisele Manciya, Harry Pangani.

**Methodology:** Zibusiso Ndlovu, Emmanuel Fajardo.

**Project administration:** Zibusiso Ndlovu, Lamin Massaquoi, Ndim Eugene Bangwen, John N. Batumba, Rachelle U. Bora, Joelle Mbuaya, Roger Nzadi, Nadine Ntabugi, Patrick Kisaka, Gisele Manciya, Ramzia Moudashirou, Harry Pangani, Patrick Mangochi, Kuziwa Kuwenyi, Reinaldo Ortuno, Douglas Mangwanya, Edmore Zvidzai, Tapiwa Mupepe, Sekesai Zinyowera.

**Resources:** Zibusiso Ndlovu.

**Supervision:** Zibusiso Ndlovu, Lamin Massaquoi, Ndim Eugene Bangwen, John N. Batumba, Roger Nzadi, Gisele Manciya, Harry Pangani, David Van Laeken, Claude Kwitonda, Yuster Ronoh.

**Validation:** Zibusiso Ndlovu.

**Visualization:** Zibusiso Ndlovu.

**Writing – original draft:** Zibusiso Ndlovu.

**Writing – review & editing:** Zibusiso Ndlovu, Lamin Massaquoi, Ndim Eugene Bangwen, Emmanuel Fajardo, Tom Ellman.

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
