## [Decision Letter · Decision Letter 0]

24 Dec 2019

PONE-D-19-30572

Diagnostic performance and usability of the VISITECT CD4 semi-quantitative test for advanced HIV disease screening

PLOS ONE

Dear Mr Ndlovu,

Thank you for submitting your manuscript to PLOS ONE. After careful consideration, we feel that it has merit but requires some revision as per reviewers' comments. Therefore, we invite you to submit a revised version of the manuscript that addresses the points raised during the review process.

We would appreciate receiving your revised manuscript by Feb 07 2020 11:59PM. To enhance the reproducibility of your results, we recommend that if applicable you deposit your laboratory protocols in protocols.io, where a protocol can be assigned its own identifier (DOI) such that it can be cited independently in the future. For instructions see: http://journals.plos.org/plosone/s/submission-guidelines#loc-laboratory-protocols

We look forward to receiving your revised manuscript.

Kind regards,

Bharat S. Parekh, Ph.D.

Academic Editor

PLOS ONE

Journal Requirements:

Reviewers' comments:

Reviewer's Responses to Questions

**Comments to the Author**

1. Is the manuscript technically sound, and do the data support the conclusions?

Reviewer #1: Yes

Reviewer #2: Yes

2. Has the statistical analysis been performed appropriately and rigorously? 

Reviewer #1: Yes

Reviewer #2: Yes

3. Have the authors made all data underlying the findings in their manuscript fully available?

Reviewer #1: Yes

Reviewer #2: Yes

4. Is the manuscript presented in an intelligible fashion and written in standard English?

Reviewer #1: Yes

Reviewer #2: Yes

5. Review Comments to the Author

Reviewer #1: 1) I suggest including your reference method in the abstract.

2) On line 176 of the methods section, the following sentence was cut-off “The Visitect CD4 LFA test kit can be stored”

3) When assessing the performance of Visitect CD4 LFA at different reference test cut-offs, I recommend including the number of samples compared at each cut-off.

4) On line 223, the authors state “In restricting the analysis to samples with CD4 < 100 cellsmm3 cut-off in the reference test, the sensitivity of the Visitect CD4 LFA was 98% [IQR: 93.1 – 99.8] and specificity was 66.6% [IQR: 62.2 – 70.8].” If the analysis was truly restricted to reference values < 100 cells/mm3 then there would be no false-positives from the test assay because there would be no reference values > 200 cells/ml. Was this analysis instead performed by changing the cut-off of the reference assay? I recommend a better explanation of this analysis here or in the methods section.

5) The authors shared the Kappa statistic of the Visitect CD4 LFA compared to the Pima CD4, but not the specificity or sensitivity. I recommend sharing the sensitivity and specificity with the Pima CD4 as the reference assay, or an explanation of why the authors chose to use venous blood tested on the BD FACScount as the reference assay for FP samples tested on the Visitect CD4 LFA.

Reviewer #2: A few suggestions to strengthen the manuscript:

1. The authors use sensitivity and specificity to describe the correlation between the different methods at the threshold of 200, which introduces some confusion as compared with test evaluations that compare positives and negatives as is done in most method comparisons. It might be more clear to describe the relationship as a correlation or concordance of methods at >200 or <200. Alternatively, it could be stated as the sensitivity or accuracy to detect <200 and the sensitivity or accuracy to detect >200.

2.Precision studies: Were any samples used that produced an equal result intensity with the 200 line? That is, precision studies should include a sample near the cutoff where changes could be realized. The nature of the samples used is not described in the Methods Section.

3. Reproducibility study: What is described is not a reproducibility study; it describes an experiment where incubation times are varied and the effect on results is observed. The authors should consider to remove this because it is small in scope, not relevant for the user because they are supposed to follow instructions, and it might give the impression that incubation times can be varied without having an effect on test results. If it were to be included, then perhaps there should be other investigations also (e.g., varying the number of drops of buffer, using partial volumes of sample, etc.) to be more complete. A true reproducibility study shows how the test performs with multiple testing of the same samples over hours, days, or longer, and with varying conditions (different users, different temperatures on days, etc.).

4. Line 176 is a partial sentence.

5. Line 253 - the ambient temperature should be stated (in some resource limited countries it could be 35C). Stability could also mean how the test performs after storage at different temperatures over time. Therefore, it should refer to "stability of the observed results".

6. In the early years of 2000, there was a method called TRAxCD4 (T Cell Dxs/Innogenetics) that was an ELISA based method that measured soluble CD4 protein in a sandwich configuration (similar to the principle in this LFA). The authors should review the literature and include any references in their manuscript.

6. PLOS authors have the option to publish the peer review history of their article (what does this mean?). If published, this will include your full peer review and any attached files.

Reviewer #1: No

Reviewer #2: Yes: Niel Constantine, Ph.D.

---

## [Author Response · Author response to Decision Letter 0]

13 Jan 2020

We would like to thank the reviewers for their time and concise inputs. 

This study was initially submitted to PLOS-One with results from two study sites (Malawi and DRC); however, the study in Zimbabwe has also been finished recently and we have herein added the results from Zimbabwe for 118 patients tested in Phase 1. We also added the co-investigator details from Zimbabwe and also added the Ethical Approvals obtained from the Zimbabwe Medical Research Council. 

Responses to reviewer comments

Reviewer #1: 

Reviewer comment 1: I suggest including your reference method in the abstract.

Author response: line 66 and 67 of the abstract do mention the reference test method used in the study (i.e BD FACSCount assay for CD4 testing)

Reviewer comment 2: On line 176 of the methods section, the following sentence was cut-off “The Visitect CD4 LFA test kit can be stored”

Author response: We have corrected the sentence and it now reads; “The Visitect CD4 LFA test kit can be stored within temperature ranges of 2 - 30oC” (line 195).

Reviewer comment 3: When assessing the performance of Visitect CD4 LFA at different reference test cut-offs, I recommend including the number of samples compared at each cut-off.

Author response: We have added the number and percentage of samples for each of the cut-off ranges (line 248, table 3)

Reviewer comment 4: On line 223, the authors state “In restricting the analysis to samples with CD4 < 100 cellsmm3 cut-off in the reference test, the sensitivity of the Visitect CD4 LFA was 98% [IQR: 93.1 – 99.8] and specificity was 66.6% [IQR: 62.2 – 70.8].” If the analysis was truly restricted to reference values < 100 cells/mm3 then there would be no false-positives from the test assay because there would be no reference values > 200 cells/ml. Was this analysis instead performed by changing the cut-off of the reference assay? I recommend a better explanation of this analysis here or in the methods section.

Author response: 

We have modified the sentence to reflect that the sensitivity/specificity of Visitect CD4 LFA at the different cut-offs is purely based on the reference assay cut-offs, whilst the Visitect CD4 LFA remains with its standard threshold of 200cells/mm3. Line 242-244. Furthermore, we avoid mentioning the words ‘false positive/false negative’ during the interpretation of the results of this restriction analysis. 

Reviewer comment 5: The authors shared the Kappa statistic of the Visitect CD4 LFA compared to the Pima CD4, but not the specificity or sensitivity. I recommend sharing the sensitivity and specificity with the Pima CD4 as the reference assay, or an explanation of why the authors chose to use venous blood tested on the BD FACScount as the reference assay for FP samples tested on the Visitect CD4 LFA.

Author response: we used the Kappa statistic to compare diagnostic performance of Visitect CD4 LFA to PIMA (and not the sensitivity/specificity) because PIMA is not a reference standard method for CD4 cell count testing, as such Kappa statistic is the most correct/recommended way to assess the agreement between two methods (where there is no reference/gold standard) as both are considered imperfect tests. 

Reviewer #2: 

Reviewer comment 1: The authors use sensitivity and specificity to describe the correlation between the different methods at the threshold of 200, which introduces some confusion as compared with test evaluations that compare positives and negatives as is done in most method comparisons. It might be more clear to describe the relationship as a correlation or concordance of methods at >200 or <200. Alternatively, it could be stated as the sensitivity or accuracy to detect <200 and the sensitivity or accuracy to detect >200.

Author response: Yes, we understand the potential ‘confusion’ that could arise with interpreting results for such a binary-test which doesn’t necessarily provide the traditional ‘positive’ or ‘negative’ results. The consequence is that Visitect CD4 LFA results that are <200 can loosely be called ‘positive’ whilst those >200 can be said to be ‘negative’. 

In our study and in data analysis packages, as long as diagnostic result outcomes are dichotomous/binary, they are automatically labeled as ‘normal’ and ‘abnormal’, which enables determination of sensitivity/specificity to a reference standard method. This is regardless of whether the result is positive/negative or is <200/>200. Nonetheless, in the manuscript, we do emphasize (in multiple occasions) that the calculated outcomes of performance are sensitivity/specificity to detect CD4 cell counts at the 200cells/mm3 threshold (meaning either >200 or less than 200), line 230-232.

Reviewer comment 2: Precision studies: Were any samples used that produced an equal result intensity with the 200 line? That is, precision studies should include a sample near the cutoff where changes could be realized. The nature of the samples used is not described in the Methods Section.

Author response: Line 199-201 in Methodology section does partly mention the strategy for assessing the precision of the index test. In line 250-253 (results section), we do mention the outcomes of precision assessment, and the fact that we used 12 samples and 2 had CD4 cell counts close to the 200cells/mm3 threshold. Nonetheless, we certainly do acknowledge the need to describe the samples used for precision; however, since we used a range of samples (in terms of CD4 cell counts) we could not manage to describe all the sample types used within the methodology/results section. 

Reviewer comment 3: Reproducibility study: What is described is not a reproducibility study; it describes an experiment where incubation times are varied and the effect on results is observed. The authors should consider to remove this because it is small in scope, not relevant for the user because they are supposed to follow instructions, and it might give the impression that incubation times can be varied without having an effect on test results. If it were to be included, then perhaps there should be other investigations also (e.g., varying the number of drops of buffer, using partial volumes of sample, etc.) to be more complete. A true reproducibility study shows how the test performs with multiple testing of the same samples over hours, days, or longer, and with varying conditions (different users, different temperatures on days, etc.).

Author response: We acknowledge that our reproducibility part of the study is limited in scope and the idea behind it was purely to explore the effect of varying the incubation times only; as we pragmatically know that this will be one of the biggest independent factors to affect test results in different real user settings. Practically, we know health-care workers sometimes do not follow the instructions for use (IFU) for many POC diagnostics and with this multi-step test; we sought to show the potential consequences of not following the instructions. We do acknowledge that our findings may be taken to mean that the incubation times can be changed, but we specifically sought to provide a clue on this question that these incubation times are un-forgiving and testers must adhere to manufacture times. To simplify the text in the manuscript, we have removed part of the reproducibility data (Table 4) to ensure limited information remains; and the Discussion section (line 339-347) provide a clear emphasis on following manufacture incubation times. 

Reviewer comment 4: Line 176 is a partial sentence.

Author response: We have corrected the sentence and it now reads; “The Visitect CD4 LFA test kit can be stored within temperature ranges of 2 - 30oC” (line 195).

Reviewer comment 5: Line 253 - the ambient temperature should be stated (in some resource limited countries it could be 35C). Stability could also mean how the test performs after storage at different temperatures over time. Therefore, it should refer to "stability of the observed results".

Author response: we have modified this and clarified that the ambient laboratory temperature range was 24–28oC. Line 269 Furthermore, we have added the word ‘result re-reading’ in the subtitle (line 268) to emphasize that the stability results are just from re-reading the stored test cassettes only. 

Reviewer comment 6: In the early years of 2000, there was a method called TRAxCD4 (T Cell Dxs/Innogenetics) that was an ELISA based method that measured soluble CD4 protein in a sandwich configuration (similar to the principle in this LFA). The authors should review the literature and include any references in their manuscript. 

Author response: We have done a literature review of this TRAXCD4 technology and many others during the early 2000 and we have added a reference of one of the review articles on this (reference number 7). We however, did not add the whole technology appraisal from early 2000s in detail in the manuscript as most of these technologies are obsolete. Also we wanted our Introduction Section to focus in the currently commercially available point-of-care (POC) or near-POC CD4 assays in the market (line 110-113).

---

## [Decision Letter · Decision Letter 1]

2 Mar 2020

Diagnostic performance and usability of the VISITECT CD4 semi-quantitative test for advanced HIV disease screening

PONE-D-19-30572R1

Dear Dr. Ndlovu,

We are pleased to inform you that your manuscript has been judged scientifically suitable for publication and will be formally accepted for publication once it complies with all outstanding technical requirements.

With kind regards,

Bharat S. Parekh, Ph.D.

Academic Editor

PLOS ONE

Additional Editor Comments (optional):

Thank you for addressing reviewers' comments. Manuscript is now acceptable for publication.

Reviewers' comments:

Reviewer's Responses to Questions

**Comments to the Author**

1. If the authors have adequately addressed your comments raised in a previous round of review and you feel that this manuscript is now acceptable for publication, you may indicate that here to bypass the “Comments to the Author” section, enter your conflict of interest statement in the “Confidential to Editor” section, and submit your "Accept" recommendation.

Reviewer #1: (No Response)

2. Is the manuscript technically sound, and do the data support the conclusions?

Reviewer #1: Yes

3. Has the statistical analysis been performed appropriately and rigorously? 

Reviewer #1: No

4. Have the authors made all data underlying the findings in their manuscript fully available?

Reviewer #1: No

5. Is the manuscript presented in an intelligible fashion and written in standard English?

Reviewer #1: Yes

6. Review Comments to the Author

Reviewer #1: 1) Again, on line 242, the authors state, "In restricting the analysis to samples with CD4<100cells/mm3 cut-off in the reference test, the sensitivity of the Visitect CD4 LFA (at its standard 200cells/mm3 threshold) in detecting samples with CD4<100cells/mm3 was 98.1% [IQR: 93.5 – 99.8] and specificity was 68.0% [IQR: 64.1 – 71.7]". If you truly restricted the analysis to samples with CD4 < 100 cells/mm3, the specificity would be 0% as there would be 0 samples > 100 cells/mm3 for the reference assay. I suspect the authors analyzed all samples (less than and greater than the cut-offs), however varied the reference test cut-offs. If this is true, changing this sentence to say, "In analyzing samples at varying reference test cut-offs, the sensitivity of the Visitect CD4 LFA..." would render it accurate.

7. PLOS authors have the option to publish the peer review history of their article (what does this mean?). If published, this will include your full peer review and any attached files.

Reviewer #1: No

---

## [Editor Report · Acceptance letter]

5 Mar 2020

PONE-D-19-30572R1 

Diagnostic performance and usability of the VISITECT CD4 semi-quantitative test for advanced HIV disease screening 

Dear Dr. Ndlovu:

I am pleased to inform you that your manuscript has been deemed suitable for publication in PLOS ONE. Congratulations! Your manuscript is now with our production department. 

With kind regards,

on behalf of

Dr. Bharat S. Parekh 

Academic Editor

PLOS ONE